# Material and Environmental Properties of Natural Polymers and Their Composites for Packaging Applications—A Review

**DOI:** 10.3390/polym14194033

**Published:** 2022-09-26

**Authors:** Prabaharan Graceraj Ponnusamy, Sudhagar Mani

**Affiliations:** School of Chemical, Materials and Biomedical Engineering, University of Georgia, Athens, GA 30605, USA

**Keywords:** natural polymers, nanocomposites, nanocellulose, packaging materials, tensile strength, water vapor barrier properties, environment impacts

## Abstract

The current trend of using plastic material in the manufacturing of packaging products raises serious environmental concerns due to waste disposal on land and in oceans and other environmental pollution. Natural polymers such as cellulose, starch, chitosan, and protein extracted from renewable resources are extensively explored as alternatives to plastics due to their biodegradability, biocompatibility, nontoxic properties, and abundant availability. The tensile and water vapor barrier properties and the environmental impacts of natural polymers played key roles in determining the eligibility of these materials for packaging applications. The brittle behavior and hydrophilic nature of natural polymers reduced the tensile and water vapor barrier properties. However, the addition of plasticizer, crosslinker, and reinforcement agents substantially improved the mechanical and water vapor resistance properties. The dispersion abilities and strong interfacial adhesion of nanocellulose with natural polymers improved the tensile strength and water vapor barrier properties of natural polymer-based packaging films. The maximum tensile stress of these composite films was about 38 to 200% more than that of films without reinforcement. The water vapor barrier properties of composite films also reduced up to 60% with nanocellulose reinforcement. The strong hydrogen bonding between natural polymer and nanocellulose reduced the polymer chain movement and decreased the percent elongation at break up to 100%. This review aims to present an overview of the mechanical and water vapor barrier properties of natural polymers and their composites along with the life cycle environmental impacts to elucidate their potential for packaging applications.

## 1. Introduction

Plastic materials are a vibrant part of the global business market because of their excellent functional characteristics and economic values. The global production of plastic materials reached 348 MT to meet the increasing demand from various market sectors such as packaging, building and constructions, automotive, electrical and electronics, household, agriculture, pharmaceutical, etc. [1]. The versatility of plastic materials for various applications increases the demand every year. Since 2000, plastics production has increased by approximately 3 to 4% every year [2]. The increasing plastic product manufacturing, use, and disposal resulted in serious environmental concerns due to inefficient existing recycling infrastructure. The plastic products manufacturing industries are facing tough challenges to deliver value-added products that reduce energy use and emissions during feedstock processing, plastic product manufacturing, and disposal. The feedstock materials used to produce plastic materials are synthesized from crude oil fossil resources. A huge amount of energy is consumed during crude oil and plastic feedstock material manufacturing processes. The production of polymer feedstock consumed about 2.50 to 4% of total U.S. primary energy consumption in 2008 [3]. The use of catalyst, solvent, and additive materials in monomer synthesis and plastic product manufacturing are major contributors to environmental burdens. The environment and human health are affected by the presence of catalyst residues and reactive flame retardants after plastic material processing and the thermal stabilizers used in the production of plastic products [4,5].

In addition to the environmental challenges related to polymer material production, the chemical leaching of unbound monomers and the migration of left-out catalysts, additives, and solvents into stored beverages and foods during product storage also cause serious problems [6]. All these concerns are pointed out as major health risks to humans and our entire ecosystem [7,8,9]. Furthermore, the end of life of plastic products was mismanaged over the past 50 years by poor recycling programs, its uncontrolled disposal in the environment as litter, and ending up in municipal solid waste (MSW) [10]. The present end-of-life management options such as the recovery of energy from MSW, recycling, and the reuse of plastic wastes led to a marginal reduction in landfill disposal but did not show any accelerated shift to reduce plastic wastes from our environment [11]. Almost 10% of plastic land litter is discharged into the ocean through stormwater, rivers, and wind [12,13,14]. The plastic litter in land and marine environments is fragmented into micro and nanosized debris by natural degradation. Some of the living species in marine and land habitats are ingesting the leaked plastic debris and loading harmful pollutants on their ecosystems [15,16,17]. The ingestion of micro/nano plastic fragments can cause serious health hazards such as cell death, oxidative stress, and innate immune system damage [18,19].

Other challenges related to plastic materials are their increasing demand in various product applications. It is expected that plastic material production will double by the year 2050 [20]. Around 40% of total plastic material production is used for packaging manufacturing [21]. The increasing use of plastic materials causes the depletion of fossil resources and an increase in crude oil price and only supports the linear economy of the extract–use–dispose model. In the context of challenges in plastics feedstock material processing, product manufacturing, use, end-of-life management, and increasing market demand, a new material system is envisioned as an alternative plastic material from renewable resources, supporting a circular economy model. This alternative plastic material system was planned to detach from the dependence on fossil-based resources and shift toward carbon-neutral energy resources [22]. The exploration of renewable resources for plastics materials led to the emergence of new materials called biopolymers or bioplastics. Polymers that are isolated from bio-based resources with or without biodegradation behavior and fossil-based resources with biodegradation behavior are defined as biopolymers or bioplastics [23,24]. The biopolymers are produced from their resources in two different ways, the extraction of polymer chains directly from biomass or of monomer from microorganisms or fossil resources or synthesis by polymerization. Some of the biopolymers synthesized from biomass had equivalent physical and chemical properties to that of fossil-based polymers (e.g., biopolyethylene derived from ethanol), and some exhibited unique properties as a new class of materials (e.g., polylactic acid, PLA) [25].

The biopolymers that are directly extracted from renewable resources are called natural polymers. They are receiving enormous attention due to their material characteristics, abundant availability, biocompatibility, and biodegradability properties. The tensile and water vapor barrier properties of biopolymers were evaluated to use them as potential alternative packaging materials [26]. The tensile and barrier properties of biopolymers determined their ability to withstand different loading and environmental conditions after being manufactured as packaging products [27,28]. The flexible films and coatings for general purpose and food packaging applications were produced from natural polymers and were suggested as a suitable alternate environment-friendly material [29,30]. The greenhouse gas (GHG) emissions and environmental impacts such as the acidification of soil, ozone depletion, and eutrophication during their cradle-to-grave or cradle-to-gate life cycle stages were also assessed using the life cycle assessment (LCA) tool.

This review article discusses the different types of natural polymers and their material properties reported in recent peer-reviewed articles. It also briefly presents the mechanical and barrier properties of natural polymers and their composites for packaging film and coating applications. In addition, the life cycle environmental impacts of natural polymers are discussed to elucidate the environmental advantages of natural polymers over fossil-based polymers.

## 2. Natural Polymers

Repeated and arranged polymer molecular units are present in some animal and plant biomass. They behave as fossil-based polymeric materials and are called natural polymers. The natural polymers that were extracted from different biomass resources are classified based on the resources used, as depicted in Figure 1 [31,32,33,34]. They are present in animals and plants as protein macromolecules of amino acids bonded by peptides or as polysaccharide macromolecules of monosaccharides bonded by glycosidic bonds or as lipid–long chain hydrocarbon molecules containing a carboxylic acid moiety.

Natural polymers are abundantly available in many renewable resources. At present, biomass resources are mainly utilized for the production of various food products, oil, feed grains, bioenergy, and cosmetic products. The production and utilization details of various resources used for the manufacture of natural polymers in the U.S.A. are presented in Table 1.

### 2.1. Protein-Based Natural Polymers

#### 2.1.1. Proteins-Based Natural Polymers from an Animal Resource

Milk is a colloidal solution constituted of fat, minerals, vitamins, and a heterogeneous mixture of the proteins casein (80%) and whey (20%). Flexible and transparent films were produced from casein and whey proteins present in the milk. The whey proteins are the aggregate of soluble globular proteins in serum albumin [46]. The casein proteins contain four forms of protein, namely, α_s1_, α_s2_, β, and κ casein (Figure 2) [47]. The whey and casein proteins are polymerized from milk by acidification and heat treatment processes and are separated by micro- and ultra-filtration techniques [48,49]. The protein molecules tend to form films due to bonding and electrostatic interaction. The film and coating properties of casein are determined from the calcium micelles formed by the hydrophobic and electrostatic interactions of protein molecules and calcium bridging elements [50]. As the native milk protein films are brittle, plasticizers were added to weaken the bonding between protein chains. The crosslinking agents and plasticizers enhanced the mechanical and physical properties of the films [51,52].

Collagen is the most abundant (about 25%) protein which is present in the cell walls of vertebrates and invertebrates [53,54]. Gelatin proteins are extracted from collagen by acetic acid hydrolysis (Figure 3). It exhibits good solubility in water. The gelatin proteins are a mixture of long and short amino acids connected by peptide bonds. The amino acid sequences determine the polymer structure and the properties of protein polymer [55]. Gelatin-based edible films and coatings were developed to use in food packaging. A gelatin polymer from fish skin was extracted with an acid-and-base treatment [56]. The improvement in the physical and mechanical properties of gelatin-based films was observed with the addition of antimicrobial, antioxidant, and lipid agents. The gelatin films and coatings were produced by dip coating, casting, and extruding [57].

Sericin is a protein extracted from silk fibers by a degumming process using boiled water. Sericin has different amino acids such as serine, glycine, glutamate, and threonine [58]. The carboxyl, amino, and hydroxyl groups are the major polar groups present in this protein. These polar groups are reactive elements that enable crosslinking between molecular chains. As the standalone film-forming characteristics of sericin are not good, it is used with other polymers to make packaging film and coatings [59,60].

The animal proteins have good film-forming abilities, with poor tensile and water vapor barrier properties. Crosslinking and plasticizers were used to increase the tensile strength of the films. They were found suitable for edible coating and films. The water vapor transmission was increased by 100% with increased pore size by crosslinking [61]. The mechanical and barrier properties of animal protein-based natural polymers are presented in Table 2.

#### 2.1.2. Proteins-Based Natural Polymers from Plant Resources

Wheat grains have starch, lipids, and gluten proteins. The gluten proteins are constituted with high contents of gliadins and glutenin bonded by disulfide, hydrogen, and ionic and hydrophobic bonds. These proteins are especially characterized by their protein molecular weights and are extracted from wheat by treatment with ethanol [65,66]. Gluten-based films for packaging applications were developed with plasticizers such as glycerol and sorbitol for the improvement of tensile properties. The tensile strength of gluten-based films is less than that of polyethylene-based materials, although the percentage of elongation is comparable with polyethylene-based materials [67,68].

Soy protein isolate (SPI), which contains 92%, protein, is extracted from soybean by removing fats, carbohydrates, fibers, and moisture. The SPI is a mixture of albumins and globulins proteins with many functional groups such as carboxyl, amine, and hydroxyls. SPI is extracted from de-fatted soy flakes by treating with either water or mild alkali (pH 7–9) at 50–55 °C and precipitated by adjusting the pH to ∼4.5 with food-grade acid [69]. The tensile properties of SPI materials were modified with plasticizer and formed into films by casting or melt processing.

Zein protein is extracted from corn by treating it with aqueous ethanol extract and a dry milling process. It contains mostly α-zein, which can self-assemble into a microstructure to form a film or coating [70,71,72]. The films formed with native zein proteins are brittle and sensitive to high relative humidity.

The plant-based protein natural polymers exhibited excellent film-forming abilities. Their brittle nature and poor resistance to moisture absorbance are the limiting factors that prevent them from being considered for packaging applications. The summary of mechanical and barrier properties of plant protein-based natural polymers is listed in Table 3.

### 2.2. Polysaccharide-Based Natural Polymers

#### 2.2.1. Polysaccharide-Based Natural Polymers from an Animal Resource

Chitin is a polysaccharide extracted from crab and shrimp shells by demineralization and deproteination processes as shown in Figure 4a. The monosaccharide units of chitin have an acetyl amine group (-CH_3_-CO-NH) and are linked by β-(1 → 4) covalent bonds [76]. The acetyl amine group present in the chitin causes strong hydrogen bonding between adjacent polymers. The antibacterial and antifungal properties and abundant availability of chitin attracted food packaging applications [77,78,79]. The tensile strength of around 18 MPa and the percentage of elongation of 6% were achievable for films manufactured from chitin natural polymer by film casting [80].

Chitosan is the natural polymer manufactured from chitin by deacetylation with base agents as shown in Figure 4a. They are available in a different range of molecular weights and degrees of deacetylation. The primary functional groups available in these polymers are hydroxyl (OH), amine (NH_2_), and ether (C-O-C) [81]. The presence of amino groups makes chitosan a positively charged polysaccharide. Chitosan is not soluble in water but is soluble in weak acidic solutions. To develop chitosan as packaging materials, hydrophilic properties attributed to hydroxyl groups were improved by crosslinking, and the elongation at break was improved by blending with plasticizer [82]. The tensile strength and percentage of elongation at the break of the chitosan films modified by citric acid crosslinking were around 13 and 48 MPa, respectively [83].

#### 2.2.2. Polysaccharide-Based Natural Polymers from Plant Resource

Thermoplastic starch (TPS) is a polysaccharide polymer that is extracted from biomass such as corn, wheat, rice, potato, cassava root, barley, and oat as shown in Figure 4b. The structure of TPS is constituted of amylose and amylopectin macromolecules [84]. The tensile strength and percentage of elongation of native starch are 5 and 50 MPa, respectively [85]. Thermoplastic starch plasticized by polyols was investigated to use as an edible coating and packaging film. Tensile strengths of 10 to 30 MPa and percentages of elongation at break of 3 to 60% were obtained with 20 to 30% glycerol as a plasticizer in packaging film production with starch biopolymers [86,87].

Cellulose is an extract from plants and is the most abundant material on earth. It forms a polymeric structure with β-D-glucopyranose units having reactive hydroxyl groups in C2, C3, and C6 and linked by a covalent bond with acetal groups in C4 and C1 [88]. They are widely extracted from wood and plant biomass resources as shown in Figure 4c. The adjacent cellulose molecules form hydrogen bonds and make rigid structures during the film-forming process. The films with microcellulose fibrils showed 80 MPa tensile strength [89]. The nanocellulose films manufactured from different resources and different extraction processes exhibited distinct tensile properties. The softwood nanocellulose films manufactured by the tempo oxidation method had a tensile strength of 82 MPa and percentage elongation at a break of 1 [90].

### 2.3. Lipid-Based Natural Polymers

Wax-based natural polymers are used as edible films and coating [91]. The wax polymers constitute majorly long chain hydrocarbons and esters. They are insoluble in water but soluble in organic solvents. The temperature dependence of wax-based film is a limiting factor in using these films in packaging applications [92]. Similarly, the use of lacquers in packaging applications is limited to coatings on metallic surfaces to avoid harmful elements in packaging materials [93,94,95]. The hydroxyl groups of acetylated fatty acids were modified to enable crosslinking between molecules to increase the tensile strength of the coating. The tensile strength of these films was found to be 1.76 MPa [96,97].

## 3. Natural Polymer Nanocomposites

The tensile and barrier properties of natural polymer materials are the primary properties that determine the functionality of the material for development as a potential packaging material. The modulus of elasticity and percentage of elongation at break of natural polymers could not be improved to a greater extent with plasticizer and crosslinking agent to match fossil resource-based polymer materials sufficiently to suggest natural polymer materials as potential packaging materials. In this context, attempts were made to develop natural polymer composite materials by combining two or more natural polymer materials extracted from natural resources to capture the unique characteristics that were attributed to their constituent materials. The constituent materials of composites are called matrix phase and reinforcement materials. The matrix phase materials generally have higher elasticity and lower modulus, and the reinforcement materials have high load-bearing abilities [98]. These two distinct phases of composite materials account for the anisotropy properties of materials at the macro level.

The constitutive relationships of composite materials are determined by the properties of reinforcement materials such as size and dispersion [99,100]. The limitation in predicting the load response characteristics of heterogeneous composite materials and their mechanical properties restricted the design and manufacturing of composite materials and their use in different product applications. Hence, homogeneity in material characteristics was obtained by reinforcing nano-structured materials in the composite material system [101,102]. Considerable improvement in physical, thermal, and mechanical properties was obtained when reinforcing nano clay, carbon nanofiber, and carbon nanotubes (CNT) in polymer matrix [103,104,105]. However, the recent environmental concerns regarding the material development process and the demand for highly functional materials necessitated the use of reinforcement material from natural resources and led to a new class of natural polymer-based nanocomposites. Much attention was given to cellulose nanofibrils (CNF) and cellulose nanocrystals (CNC), which are abundant on earth and possess remarkable characteristics for use as reinforcement materials in natural polymer nanocomposites. Nanocellulose-based biocomposites were developed for packaging applications.

### 3.1. Protein—CNF Composite Films

The mechanical and barrier properties of protein polymers were improved by reinforcing CNF in the protein biopolymer matrix. The interfacial adhesion of CNF in protein polymer and the macromolecule structure by the hydrogen bonding network determined the composite properties. The composites were manufactured with different CNF loading percentages, and the material properties were evaluated to determine their suitability for packaging applications.

The CNF reinforcement in casein films increased the tensile strength of the films [106,107]. The maximum tensile strength (5.5 MPa) increased by 200% and the elongation at break decreased by 63% with 3% CNF reinforcement. The CNF reinforcement increased the tortuosity and reduced the diffusion of water vapor in the composites. The water vapor permeability increased for 1% CNF film and decreased beyond 1% CNF [108]. This was the same as a control for the 3% CNF reinforcement (2.7 × 10^17^ g m h^−1^ kPa^−1^ m^−^^2^). Similarly, in whey protein–CNC composite films, the tensile strength (4 MPa) and Young’s modulus (100 MPa) increased by 100 and 43%, respectively, and elongation at break (10~27%) and water vapor permeability (3.5 × 10^−5^ g m h^−1^ kPa^−1^ m^−2^) decreased by 70 and 40%, respectively, for the CNC loading of 5% [107]. The hydrophilicity of composite films increased with an increase in CNC. The CNCs were agglomerated for more than 5% of reinforcement in both casein and whey protein films.

Mondragon et al. [109] investigated composite films manufactured from gelatin biopolymer with CNF and CNC reinforcement. The tensile strength of the composite film decreased by 20% due to the poor adhesion between nanocellulose and gelatin, but the increase in CNF percentage improved the oxygen barrier property. The tensile strength, Young’s modulus, percentage of elongation, water vapor transmission rate, and oxygen transmission rate at 5% CNF loading were 100 MPa, 5 GPa, 5%, 97 g mm/m^2^ day, and 0.15 cc mm/m^2^ day, respectively [109].

As the native sericin films had poor tensile strength, it was reinforced with CNF to manufacture films. The glycerol was added as a plasticizer during the film casting. The agglomeration of CNF was observed beyond 10% CNF reinforcement. The hydroxyl groups present in CNF increased the hydrophilic nature of the sericin–nanocellulose composite films. However, due to the reaction between the hydroxyl and carboxyl groups of CNFs and sericin, stable water solubility was observed in the sericin–CNF composite films. The tensile strength, Young’s modulus, and elongation at break of composite films at 10% CNF reinforcement were 28.20 MPa, 805.96 GPa, and 5%, respectively [110].

Gluten–nanocellulose composite cast films were manufactured with carboxylate nanocellulose by Rafieian et al. [111]. To avoid the agglomeration of CNF during composite film making, the hydroxyl groups of CNFs were replaced with the carboxyl group. The tensile strength of the films was increased by 60% compared with the native gluten films. The maximum tensile strength of 5.4 MPa and elongation at break of 285% were reported at 7.5 wt.% CNF loading in the composite film. The hydrophilicity of films increased when increasing CNF up to 7.5 wt.% loadings [111].

The development of eco-friendly composites from soy protein isolate (SPI) and nanocellulose extracted from cotton and licorice was explored in two different studies by Han et al. and Wang et al. [112,113]. The SPI composites with CNC extracted from cotton had a tensile strength of 31.19 MPa, a percentage of elongation at break of 17, and a Young’s modulus of 1023 MPa at 20% CNC loading. The maximum tensile stress and Young’s modulus were about 100% more than that of the films without CNC reinforcement. The present elongation was about 80% less than that for the control films without CNC reinforcement [112]. In contrast, the composites with CNF from licorice obtained the maximum tensile strength and percentage elongation of 11.17 MPa and 63.80, respectively, at 6% CNF loading. The tensile strength was about 38% more and the percent elongation at break was 36% less than those for the film without CNF reinforcement. The intermolecular hydrogen bonds between SPI and CNF improved water resistance and thermal stability with this composite.

### 3.2. Polysaccharides–CNF/CNC Composite Films

Chitosan and starch polysaccharides were reinforced with CNF and CNC to produce composite materials. The chitosan–CNF composite film with 15% CNF and 18% glycerol resulted in tensile strength of 52.70 MPa, percent elongation at break of 10.30, Young’s modulus of 1367.90 MPa, and water vapor permeability of 5.25 × 10^−4^ g m h^−1^ kPa^−1^ m^−2^ [114]. The tensile properties of the film were comparable with LDPE and PP polymer films. It was found that the thermal, barrier, microbial, and degradation properties of the chitosan biopolymers could be improved by reinforcement with CNC. The maximum tensile strength of 120 Mpa and percentage of elongation at break of 6 were achieved at 20% CNC. The maximum tensile strength and percent elongation at break were about 42% more and 4 times less than the films without reinforcement. The water uptake was also reduced by 50% at 20% CNC reinforcement [115]. As the chitosan–nanocellulose composites have dramatically tunable mechanical, thermal, barrier, microbial, and degradation characteristics, they emerge as the best candidates for composite materials for packaging applications.

Starch-based biopolymer composite films were produced by reinforcing CNF and CNC extracted from cotton fibers, sugar beet pulp, and other biomass resources. The glycerol was used as a plasticizer during the film casting to reduce the brittleness of films. The CNF reinforcement increased the maximum tensile stress between 46 to 68% and decreased the percent elongation at a break between 35 to 102%. The water vapor permeability was less than 60% when compared with the films without CNF reinforcement. The agglomeration of CNF and CNC limited the reinforcement loading beyond 20% and 30%, respectively. The starch–CNF composite films showed improvement in hydrophobicity and oxygen transmission rate compared with native starch films [116,117,118].

## 4. Environmental Impact Assessment of Natural Polymers and Their Nanocomposites

The motivating factor for considering natural polymers as emerging alternatives to fossil-based materials in packaging applications is their comparable or tunable mechanical and barrier properties. It is evident from the reported tensile and barrier properties that the natural polymers have comparable or tunable mechanical properties with fossil-based polymer materials such as LDPE, PP, and PS and could be used in products that require structural performance manufactured from fossil-based materials. Similarly, the natural polymer–CNF composites have comparable tensile strength with LDPE–carbon nanofiber composites [119] and could be used as replacement materials for that class of polymer nanocomposite materials.

Apart from the evaluation of material performance characteristics, the assessment of the environmental impacts of material is also required before recommending this class of natural polymers as packaging materials [120]. The choice of natural resources for the feedstock of alternative polymer materials gave the advantage of achieving carbon neutrality [121]. In addition, the environmental credentials of natural polymer materials have to be evaluated by assessing their manufacturing processes to suggest them as alternative materials to fossil fuel-based material for packaging applications. Many tools are available for analyzing the environmental impacts of materials and processes. Some of the important tools used for effective analysis are life cycle assessment (LCA), environmental impact analysis (EIA), material flow analysis (MFA), and ecological footprint (EF) [122]. Out of all the tools, LCA is at the forefront due to its extraordinary capability in evaluating the environmental impact of a product from cradle to grave through a systematic approach.

### A Generic Approach for LCA of Natural Polymer Nanocomposite Packaging Product

The LCA for a packaging product that is made of natural polymer and its nanocomposites can be performed by following ISO framework guidelines. As natural polymers are proposed as alternatives to fossil-based materials, the comparative LCA approach can be set as a goal for environmental impact investigation. The scope and boundary for LCA can be described from the packaging product model and product life cycle stages [123]. The life cycle of the product starts from the biomass and passes through different stages of the conversion process. A typical nanocomposite packaging product will follow four distinct stages during its life cycle:Natural polymer matrix and reinforcement materials production.Natural polymer nanocomposite production.Packaging product manufacturing.Packaging product end-of-life management.

The first two stages of the product life cycle are associated with material development processes for the packaging products. The third stage is related to packaging product manufacturing, and the fourth stage is associated with recycling and disposal activities. The system boundaries for the LCA of a packaging product can be referred to as any one of the processes between cradle and gate or may encompass the whole life cycle from cradle to grave. The life cycle inventory datasets are compiled and evaluated within these system boundaries to find out the potential impacts of a packaging product on the environment.

## 5. Overview of LCA Studies on Natural Polymers and Their Nanocomposites

### 5.1. LCA of CNF and CNC Natural Polymer Manufacturing 

The high mechanical strength, renewability, abundance, biodegradability, and biocompatibility of CNF and CNC make them attractive for use as natural polymer reinforcement materials. CNF and CNC are extracted from plant-based biomass resources through chemical, enzymatic, and/or mechanical treatment. Nanoscale fibrillation by chemical pretreatment and mechanical treatment consumed a great amount of energy and chemicals and limited the use of CNF/CNC in various product applications. Hence, the environmental impact of manufacturing CNF/CNC was studied with a LCA. As CNF and CNC are bio-based biodegradable natural polymers and are found in a wide range of applications such as packaging products, pharmaceuticals, hydrogels, aerogels, electronics products, and biomedicals, the system boundaries for LCA studies were considered to be from the cradle to the gate. The material resources, background data, process conditions, and methodologies used in the LCA studies related to CNF/CNC manufacturing processes are briefed and presented in Table 4.

Bleached and unbleached sulfate pulps were considered as feedstock for the production of CNF/CNC. Enzymatic and chemical processes (tempo, chloroacetic acid etherification, and carboxymethylation) were used for pretreatment, and microfluidizer, homogenizer, and sonication were used as the main treatments. The enzymatic pretreatment process had a lower environmental impact (energy use: 15 MJ/kg and GWP: 0.30 kg CO_2_ eq./kg), and chloroacetic acid etherification had a higher environmental impact (Energy use: 5440 MJ/kg and GWP: ~300 kg CO_2_ eq./kg). The higher environmental impact of the chloroacetic acid pretreatment process is due to the consumption of solvent (Ethanol: 26 kg/kg of CNF and Isopropanol: 44 kg/kg of CNF) made from fossil resources. In the case of the main treatment processes, the environmental impact of the sonication process had the highest energy use: 12,170 MJ/kg and GWP of ~800 kg CO_2_ eq./kg) [124]. The energy use and GWP of other chemical and enzymatic pretreatment processes and mechanical main treatment processes used in the manufacture of CNF are presented in Figure 5 and Figure 6 [124,125]. The energy demand and GWP of CNF/CNC manufacturing processes with enzymatic pretreatment and microfluidizer and homogenizer main treatment were also lower than with the carbon nanofiber production processes [125,128,129] (Figure 7 and Figure 8).

### 5.2. LCA of Chitosan Natural Polymer Manufacturing Process

Muñoz et al. [42] conducted an LCA of chitosan and chitin manufactured from snow crab and shrimp shells from a consequential LCA perspective. Two chitosan manufacturing processes were studied. The first process used snow crab shell waste that was processed in Canada and exported to China for chitin manufacturing. The chitin was further processed in Europe for the manufacturing of chitosan. The protein sludge generated during chitin manufacturing was used as animal feed. The wastewater and NaOH waste generated during the chitosan production were treated in Europe. In contrast, in the second type of chitosan manufacturing, shrimp shell waste was used as the resource material and was manufactured in India. The wastewater generated in this process was treated and discharged into the sea. The extracted protein sludge was recycled as fertilizer, and calcium salts were disposed of as inert landfill waste. The impact was assessed using International Life Cycle Data System (ILCD) midpoint impact assessment. A short outline of impact assessment results for chitosan is presented in Table 5 [42].

### 5.3. LCA of the Thermoplastic Starch Polymer Manufacturing Process

Dinkel et al. [130] performed an LCA of thermoplastic starch (TPS) polymers to compare the results with fossil-based low-density polyethylene (LDPE) polymers. The scope of the LCA study was focused on starch polymers that were extracted from potatoes and maize and were disposed of via two waste management processes, incineration and landfilling or composting. The system boundary for the study was considered to be the cradle to the gate. It was assumed that 80% of LDPE waste would be incinerated and 20% of waste would be landfilled. It was concluded from the LCA study that TPS had better performance than LDPE in energy demand, GHG emissions, human toxicity, and salinization except for eutrophication.

Dinkel et al. [130] also conducted an LCA of films (area—100 m^2^, thickness—150 µm) made from TPS and compared the results with LDPE. This investigation was performed for the conditions applicable to Switzerland with the assumption that 20% of waste would be landfilled and 80% would be incinerated. The LCA results for TPS and LDPE films, at area of 100 m^2^ and thickness or 150 µm, are given in Table 6 [130,131].

## 6. Current Knowledge on and Opportunities for Packaging Films Made of Natural Polymers and Their Composites

Proteins and polysaccharides have several functional groups in their polymer structures. The hydroxyl, carboxyl, amino, and thiol functional groups of natural polymers were modified to enhance material properties. The surface-modified natural polymers behaved like hybrid materials and showed improved adhesion, wettability, and mechanical and hydrophobic properties. Because of their nontoxic, antioxidant, antibiotic, antimicrobial, and biocompatible nature, natural polymers are widely used in the development of drug delivery, wound dressing, organ implant, and tissue engineering materials for pharmaceutical and biomedical industrial applications. The food, textiles, water treatment, and cosmetics products in the present market contain natural polymers as emulsifiers and as antimicrobial and thickening agents. The film-forming abilities and bonding properties of natural polymers increased their uses in packaging and coating applications [132,133,134].

The packaging films were manufactured using natural polymers and their composites and investigated to demonstrate their capabilities and suggest potential packaging materials. The brittleness and hydrophilicity of the biopolymers were identified as the major challenges that would limit their use for packaging applications. Crosslinkers and plasticizers such as glycerin, glycerol, and sorbitol helped to some extent to improve the mechanical and barrier properties. They reduced the chain-to-chain interaction and increased the flexibility and resistance to fracture [135]. The cross-linked casein films exhibited comparable tensile properties with LDPE films. Similarly, the tensile properties of chitosan and starch were improved by plasticizer to match the tensile properties of LDPE films. The maximum tensile stress, Young’s modulus, and percent elongation at the break of the protein and polysaccharide films were reported between 5 and 50 MPa, 400 and 1400 MPa, and 3 and 350, respectively. Another challenging aspect of packaging film is barrier properties. Films made of SPI and zein have lower tensile strength and had comparable barrier properties with LDPE films. The hygroscopic nature of the gelatin biopolymer increased the swelling of the film while in contact with high moisture content and restricted its use as packaging material. The investigations of tensile and barrier properties of the natural polymers showed that the natural polymers have limitations in use as alternative materials to general-purpose flexible packaging films. Instead, it has the potential for use as edible packaging films and coatings. The poor resistance to water contact and water vapor transmission are the key research challenges that need to be addressed to use natural polymers as potential alternate materials for packaging applications.

To improve the mechanical and barrier properties of natural polymer films, nanocellulose-reinforced natural polymer composites were developed. The CNF and CNC are compatible with all the natural polymers. The tensile strength increased by 200, 100, 60, 100, 42, and 35% with reinforcement with CNF/CNC with casein, whey, gluten, soy, chitosan, and starch natural polymers, respectively. The CNF/CNC showed good dispersion ability and interfacial adhesion with natural polymers. The hydrogen bonding networks due to the interaction of CNF/CNC with natural polymer improved the tensile strength of the films. The increased interaction of biopolymer and reinforcement materials reduced the movement of the polymer chain and reduced the percent elongation at the break between 35 to 100%. The water vapor permeability decreased between 40 to 60% in whey protein, chitosan, and starch-based natural polymer composite films. The presence of CNC or CNF in natural polymer prevented the diffusion of water vapor and increased the barrier properties. The maximum CNF/CNC weight loading used for the manufacture of nanocomposites was from 5 to 30%; when the CNF/CNC reinforcement increased beyond the limit, they agglomerated in the composite structure and reduced the tensile and barrier properties. Potential research opportunities are available for the improvement of thermal stability, CNF/CNC dispersion ability, and composite development with surface-modified nanocellulose reinforcement with gluten, SPI, and chitosan natural polymer matrix for the improvement of barrier properties.

The reactive hydroxyl and carboxyl groups in natural polymers exhibited hydrophilic behavior and caused poor structural stability when exposed to moisture or high humidity. The hydrophilic groups were replaced with hydrophobic groups by crosslinking with chemicals such as acetic anhydride, capric acid, lauric acid, and stearic acid to improve hydrophobicity [136,137,138]. The surface-modified molecules decreased the dispersion stability and increased the agglomeration during film production or coating applications [139]. The polymer chains with low cohesive strength and many branches or rigid structures led to surface segregation during solidification and drying [140,141,142,143]. The crosslinking of natural polymers with glutaraldehyde, citric acid, adipic acid, and oxalic acid improved the barrier, antimicrobial, adsorption, and mechanical properties [144,145,146].

The environmental benefits of using renewable feedstock resources for natural polymer manufacturing processes are obtained through carbon regaining during natural polymer degradation in compost and landfills. Hence, the environmental impacts of natural polymer manufacturing processes were evaluated using the LCA method. The LCA of CNF manufacturing captured the carbon footprint and the amount of energy consumed by various extraction processes. The energy use and carbon footprint of CNF manufacturing was greatly influenced by the pretreatment and main treatment processes. The regional factors, chemical pretreatment, and main treatment by sonication also caused serious environmental impacts during CNF manufacturing. The promising future for CNF is that the energy demand with the mechanical manufacturing methods was less than the energy requirement for manufacturing other nanofiller materials such as graphene and carbon nanotubes. Future research on mechanical pretreatment and main treatment will increase the possibilities of CNF commercialization [147,148]. Further, future LCA studies on protein-based polymer material extraction processes would provide opportunities to demonstrate their capability as biodegradable packaging materials.

Polysaccharide-based natural polymers such as chitosan and starch have environmental benefits in end-of-life management scenarios. The environmental profiles of chitosan manufacturing processes were determined by the material supply chain model adapted, the environmental impact of byproducts produced, and the marginal demand for chitosan. The LCA of starch film manufacturing processes revealed that starch films had environmental advantages over LDPE film in all impact categories except water depletion and eutrophication.

The global market demand for biopolymers is projected to be about USD 10,447.2 million with a cumulative average growth rate of 17%. The environmental impacts of the waste disposal of fossil-based polymer products, enforcement of regulations, instability in crude oil price, and depletion of fossil resources are the key driving factors in the development of natural polymers for industrial applications such as packaging, medical instruments, pharmaceuticals, agriculture, cosmetics, construction, and automotive sectors. It is also forecasted that the biopolymer market in the Asia-Pacific region will grow from USD 3100 million to USD 6300 million, the highest growth rate in the world. The increasing awareness among consumers of using bio-based products and government regulations increases the use of biopolymers in the Asia-Pacific region. However, the cost of biopolymers is in the range of USD 3500 to 5200 per ton, which is much higher than that of fossil-based polymers. The high cost and marginal environmental impact of the biopolymer synthesis and extraction processes are key challenges that need to be addressed to accelerate the commercialization of biopolymer production and meet market demand [149,150].

## 7. Conclusions

The mechanical and barrier properties of natural polymers and their composites were studied to understand their performance under mechanical loading and environmental conditions. Although the use of natural polymers for packaging products is limited by material characteristics such as brittleness and a hydrophilic nature, the incorporation of plasticizer, crosslinker, and nanocellulose reinforcement helped to overcome the material properties to a great extent. The environmental burdens of manufacturing natural polymers were highlighted in LCA studies. The current knowledge from the literature on material and environmental properties revealed the capabilities of natural polymers and their nanocomposites for consideration as potential alternate packaging materials. Similarly, the LCA studies opened up avenues for improving manufacturing methods to address environmental concerns. Moreover, natural polymers and their nanocellulose composites could replace a considerable share of fossil-based packaging materials. Hence, future economic analysis of natural polymers and nanocomposites-based packaging products would strengthen the commercialization of packaging products made from natural polymers and their nanocomposites.

## Figures and Tables

**Figure 1 polymers-14-04033-f001:**
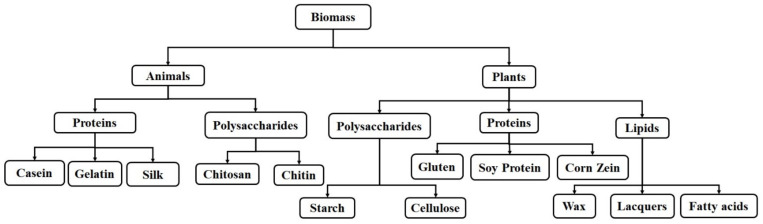
Types of natural polymers extracted from renewable resources for packaging applications.

**Figure 2 polymers-14-04033-f002:**
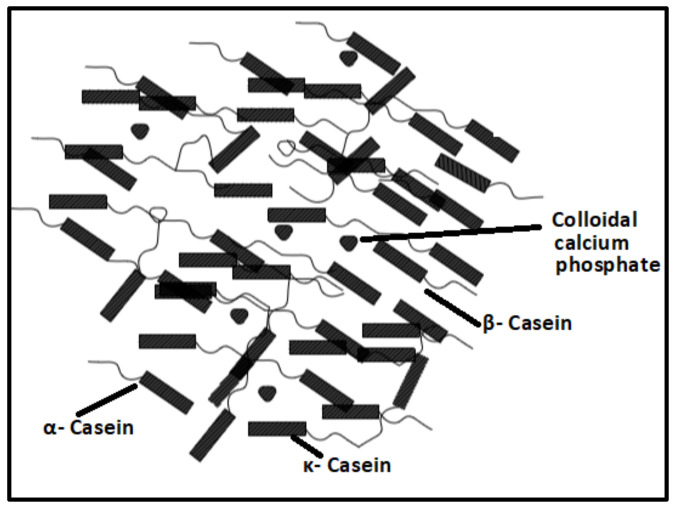
Casein polymer structure.

**Figure 3 polymers-14-04033-f003:**
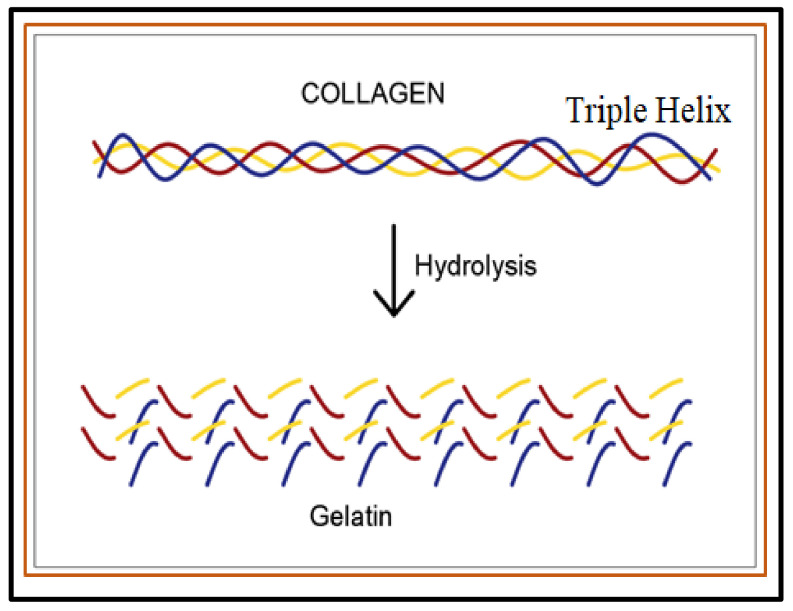
A change in collagen polymer structure during hydrolysis.

**Figure 4 polymers-14-04033-f004:**
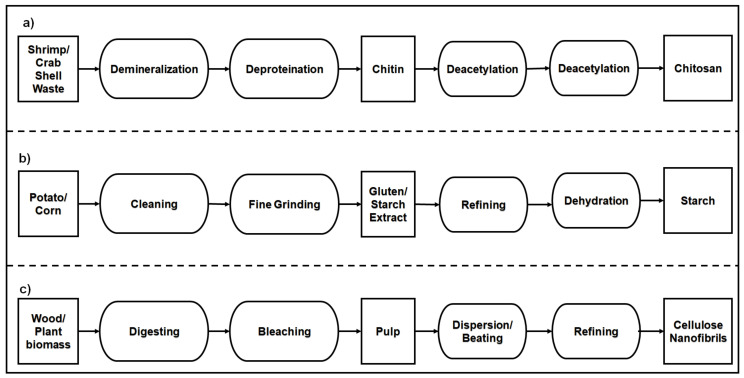
The production of polysaccharide natural polymers (**a**) Chitosan (**b**) Starch (**c**) Cellulose Nanofibrils (CNF).

**Figure 5 polymers-14-04033-f005:**
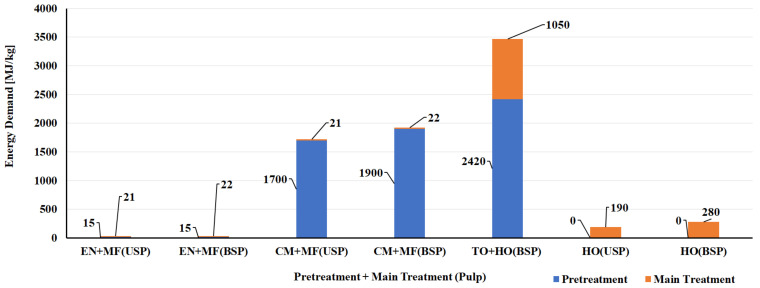
Energy demand for different pretreatment and main treatment processes of manufacturing CNF from bleached and unbleached sulfate pulp. Note: BSP—Bleached sulfate pulp, CM—Carboxymethylation pretreatment, EN—Enzymatic pretreatment, HO—High pressure homogenizer, MF—Microfluidizer, TO—Tempo oxidation pretreatment, USP—Unbleached sulfate pulp.

**Figure 6 polymers-14-04033-f006:**
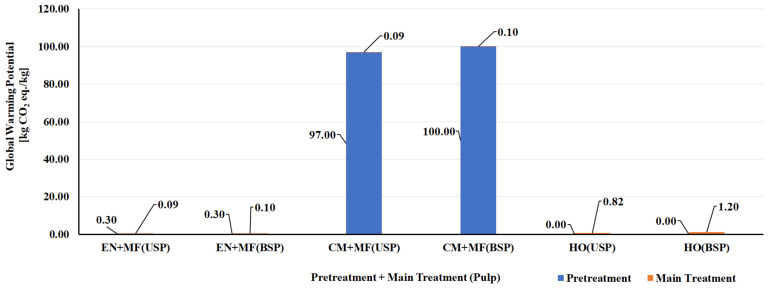
Global warming potential (GWP) of different pretreatment and main treatment processes of manufacturing CNF from bleached and unbleached sulfate pulp. Note: BSP—Bleached sulfate pulp, CM—Carboxymethylation pretreatment, EN—Enzymatic pretreatment, HO—High pressure homogenizer, MF—Microfluidizer, TO—Tempo oxidation pretreatment, USP—Unbleached sulfate pulp.

**Figure 7 polymers-14-04033-f007:**
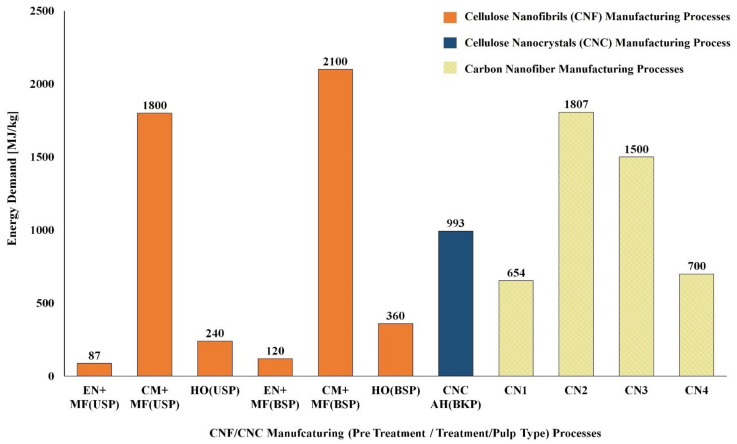
Comparison of the cumulative energy demand (CED) for CNF and CNC manufacturing processes with carbon nanofiber manufacturing processes. Note: AH—Acid hydrolysis, BKP-Bleached kraft pulp, BSP—Bleached sulfate pulp, CM—Carboxymethylation pretreatment, CN1—Carbon nanofiber from methane feedstock, CN2—Carbon nanofiber from methane feedstock with an H_2_ carrier gas, CN3—Carbon nanofiber from ethylene feedstock with an H_2_ carrier gas, CN4—Carbon nanofiber from benzene feedstock, CNC—Cellulose nanocrystals, EN—Enzymatic pretreatment, HO—High pressure homogenizer, MF—Microfluidizer, USP—Unbleached sulfate pulp.

**Figure 8 polymers-14-04033-f008:**
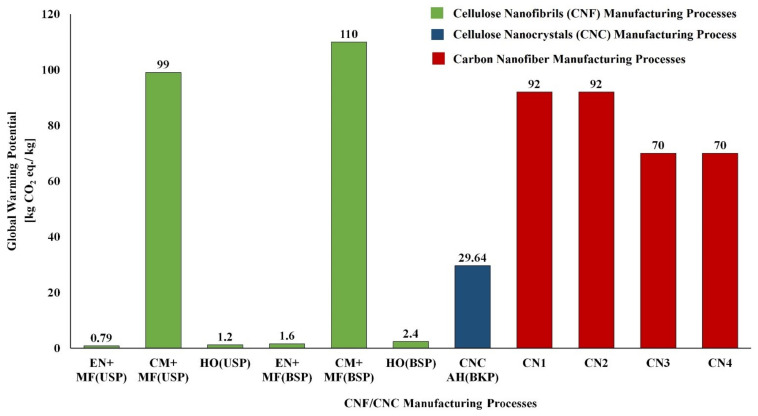
Comparison of GWP of different CNF and CNC manufacturing processes with carbon nanofiber manufacturing processes Note: AH—Acid hydrolysis, BKP-Bleached kraft pulp, BSP-Bleached sulfate pulp, CM—Carboxymethylation pretreatment, CN1—Carbon nanofiber from methane feedstock, CN2—Carbon nanofiber from methane feedstock with an H_2_ carrier gas, CN3—Carbon nanofiber from ethylene feedstock with an H_2_ carrier gas, CN4—Carbon nanofiber from benzene feedstock, CNC—Cellulose nanocrystals, EN—Enzymatic pretreatment, HO—High pressure homogenizer, MF—Microfluidizer, USP—Unbleached sulfate pulp.

**Table 1 polymers-14-04033-t001:** Renewable resources and their production volume in the U.S.A. during the year 2017.

Renewable Resources	Natural Polymer Type and Compositions	Production Volume (Million Metric Tons)	Current Use	Reference
Milk	Contains 33 g of protein/L. 80% casein and 20% whey protein	97.76	Used as a fat substitute. Butter, dry skim milk, cheese, whey, whey protein concentrate, and lactose are produced from milk.	[35,36]
Pork & Beef	More than 29% gelatin is available in pig skin. In beef meat, 10.6~21.9% of gelatin protein available in rib and shank	11.91	Used as meat. By-products such as skin, bones, and connective tissues are used to produce gelatin	[36,37]
Wheat	Contains 76.5% starch	47.38	Used for the production of food products	[36,38]
Soybeans	Contains 31.7 to 58.9% protein	120.07	Source for animal protein and vegetable oil	[36,39]
Corn grain	Contains about 70–72% starch	371.10	Source for corn meal, starch, oil, bioethanol, syrup, sugar, and feed grain	[36,40]
Potato	Contains 20% of potato dry matter with 60–80% of starch	22.91	Source for food products and starch	[36,41]
Crustaceans (Shrimp and Crab)	Crab shell contains 9.6% chitin and shrimp shell contains 4% chitin	0.32	Source for seafood and compost	[42,43]
Forestry biomass resources	40~50% cellulose	139.71	Biofuels, wood products such as timber, lumber, etc.	[44,45]
Agricultural biomass resources	25~40% cellulose	130.64	Source for bioenergy, biofuels, and bioproducts	[44,45]
Waste (Agricultural wastes, forestry wastes)	25~50% cellulose	61.69	Source for compost, bioenergy	[44,45]

**Table 2 polymers-14-04033-t002:** Tensile and barrier properties of animal protein films.

Natural Polymer	Plasticizer/ Crosslinker	TS ^a^ (MPa)	YM ^b^ (MPa)	Elongation at Break (%)	Tensile Test Conditions	WVP ^c^ × 10^20^ (g m h^−1^ kPa^−1^ m^−2^)	Reference
α_s1_—Casein Films	--	0.004	--	38	Film size 20 × 50 mm. Loading Rate—50 mm/min	--	[62]
α_s1_—Casein Films	Transglutaminase (Enzyme)	0.01	--	75		--	
α, β and κ—Casein Films ^f^	--	52 ± 0.20	1107 ± 11	8 ± 2	Loading rate—20 mm/min ^d^	--	[63]
α, β and κ—Casein Films ^g^	--	49 ± 3	1391 ± 48	6 ± 2		--	
α, β and κ—Casein Films ^h^	10% Glycerol	36 ± 0.40	693 ± 38	25 ± 9		--	
α, β and κ—Casein Films ^i^	10% Glycerol	21 ± 0.30	497 ± 40	17 ± 1		--	
α, β and κ—Casein Films ^j^	10% Glycerol	33 ± 2	765 ± 109	15 ± 7		--	
α, β and κ—Casein Films ^k^	10% Glycerol	48 ± 2	1004 ± 40	8 ± 3		--	
Whey Protein Films	40% Sorbitol	18	650	5	Loading rate—100 mm/min ^d^	--	[64]
Whey Protein Films	15% Glycerol	29	1100	4		--	
Fish Gelatin Films ^l^	10 wt.% glycerol	36.52 ± 2.98		1.79 ± 0.54	Sample size 4.75 × 22.25 mm ^e^	5.33 ± 0.16	[56]
Fish Gelatin Films ^m^	10 wt.% glycerol	43.02 ± 0.52		2.31 ± 0.33		5.47 ± 0.10	
Fish Gelatin Films ^n^	10 wt.% glycerol	52.36 ± 3.16		2.88 ± 0.68		6.52 ± 0.16	

^a^ TS—Tensile strength. ^b^ YM—Young’s modulus. ^c^ ASTM test for water vapor permeability—E96-95. ^d^ ASTM test for tensile strength—D882. ^e^ ASTM test for tensile strength—D1708-93. ^f^ Casein in NaOH/H_2_O solution and Heat treated at 130 °C/18 h. ^g^ Casein in 3-aminopropyl triethoxy silane solution—Heat treated at 130 °C/18 h. ^h^ Casein in NaOH/H_2_O solution- Air dried. ^i^ Casein in NaOH/H_2_O solution—Heat treated at 130 °C/18 h. ^j^ Casein in 3-aminopropyl triethoxy silane solution- Heat treated at 130 °C/18 h. ^k^ Casein in 3-aminopropyl triethoxy silane solution- Air dried. ^l^ Gelatin solution without pH modification. ^m^ Gelatin solution with HCl acid modified pH (2.0). ^n^ Gelatin solution with NaOH base modified pH (10.0).

**Table 3 polymers-14-04033-t003:** Tensile and barrier properties of plant protein films.

Natural Polymer	Plasticizer/Crosslinker	TS ^a^ (MPa)	YM ^b^ (MPa)	Elongation at Break (%)	Tensile Test Conditions ^c^	WVP ^d^ (g m h^−1^ kPa^−1^ m^−2^)	Reference
Glutenin-rich Wheat Gluten Film	20% Glycerol	5	--	100	Sample Size: 2.54 × 10 cm. Loading rate: 508 mm/min	6.94 × 10^4^	[73]
Gliadin-rich Wheat Gluten Film	20% Glycerol	15	--	350	--	1.11 × 10^5^	
SPI Film	50% Glycerin	2.80 ± 0.30	--	165.70 ± 15	Film size: 2.54 × 15 cm	9.66 × 10^−9^	[74]
Zein Film	--	6.70 ± 0.37	409.86 ± 7.62	1.96 ± 0.18	Film size: 40 × 10 × 0.47 ± 0.12 mm	1.69 × 10^−4^	[75]
Zein Film	10% Tributyl Citrate	17.80 ± 4.26	556.29 ± 29.42	4.53 ± 0.54		1.64 × 10^−4^	

^a^ TS—Tensile strength. ^b^ YM—Young’s Modulus. ^c^ Films conditioned in a chamber at 23 °C and 50% RH for at least 48 h. ^d^ ASTM test method for Water vapor permeability—E96-95.

**Table 4 polymers-14-04033-t004:** Resource, production process, LCI data model, and impact assessment method used in the LCA of CNF/CNC manufacturing processes.

S. No	Biomass Resource	Pretreatment Method	Main Treatment Method	LCI Resources	Life Cycle Impact Assessment	Reference
CNF Manufacturing Process						
1	Bleached kraft wood sulfate pulp	Tempo oxidation	Sonication	Literature data/SimaPro USLCI/Ecoinvent	Method: Cumulative energy demand (CED) and Eco-Indicator 99 (EI99)	[124]
2	Bleached kraft wood sulfate pulp	Tempo oxidation	Homogenization	Literature data/SimaPro USLCI/Ecoinvent	Method: CED and EI99	[124]
3	Bleached kraft wood sulfate pulp	Chloroacetic acid etherification	Sonication	Literature data/SimaPro USLCI/Ecoinvent	Method: CED and EI99	[124]
4	Bleached kraft wood sulfate pulp	Chloroacetic acid etherification	Homogenization	Literature data/SimaPro USLCI/Ecoinvent	Method: CED and EI99	[124]
5	Bleached & unbleached sulfate pulp	Enzymatic	Micro fluidizer	Process research institute, Literature & Ecoinvent database	Method: CED and ReCiPe	[125]
6	Bleached & unbleached sulfate pulp	Carboxy methylation pretreatment	Micro fluidizer	Process research institute, Literature & Ecoinvent database	Method: CED and ReCiPe	[125]
7	Bleached & unbleached sulfate pulp	No pretreatment	Homogenization	Process research institute, Literature & Ecoinvent database	Method: CED and ReCiPe	[125]
8	Carrot waste	Enzymatic	Homogenization	Laboratory scale process data & Ecoinvent	Method: CED and ReCiPe	[126]
**CNC Manufacturing Process**						
1	Unripe coconut fiber	Chopping/Washing/Bleaching	Acid Hydrolysis/Dialysis	Laboratory scale process data & Ecoinvent	Method: ReCiPe	[127]
2	Cotton fiber	Chopping	Acid Hydrolysis/Dialysis	Laboratory scale process data & Ecoinvent	Method: ReCiPe	[127]
3	Bleached kraft pulp	--	Acid Hydrolysis	US LCI and US Ecoinvent	Method: TRACI	[128]

**Table 5 polymers-14-04033-t005:** The LCA results of chitosan natural polymers [42].

Natural Polymers	Global Warming [kg CO_2_ eq.]	Acidification [mol. H+ eq.]	Eutrophication [mol. N eq.]	Ozone Depletion [kg CFC-11 eq.]	Water Depletion [m^3^]
Chitosan (Shrimp shell)	12.2	0.684	2.82	7.05 × 10^−6^	−0.236
Chitosan (Crab shell)	77.1	−0.261	3.12	1.23 × 10^−5^	5.87

**Table 6 polymers-14-04033-t006:** The LCA results for TPS and LDPE (Dinkel et al., [130] & Patel et al., [131]).

Films	Energy Demand [MJ]	Global Warming [kg CO_2_ eq.]	Acidification [kg SO_2_ eq.]	Eutrophication [kg PO_4_ eq.]	Ozone Depletion [kg ethylene eq.]	Salination [H^+^/mol]
TPS	649	25	0.24	0.13	0.1	40
LDPE	1340	67	0.24	0.02	0.18	120

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
