# Peer review of "Material and Environmental Properties of Natural Polymers and Their Composites for Packaging Applications—A Review"

_polymers, 2022, doi:10.3390/polym14194033_

Round 1

Reviewer 1 Report

The manuscript "Material and environmental properties of natural polymers and their composites for packaging applications – A review" shows the use of alternative natural polymers to replace synthetic plastics. There some good parts in the manuscript and some which needs revision. The different materials used are shown in Tables but those are not enough as the reader needs have some actual research (pictures of other work) shown how those natural plastics are processed.

1. Please include some industrial process where natural polymers either starch, cellulose or others already applied.

2. It would be good as authors mention in some parts how good some materials match properties in comparison to PP and PE. Please add some sort of comparison to see where the gap that still need to achieve replacement from synthetic to natural plastics.

3. How much of natural plastic is used and in the market? A price comparison of cost and production would be helpful.

4. There are already some natural plastics used especially in India, please add some chapter to address such as it shows that progress already occur.

5. An outlook before conclusion would be helpful as well to address the challenges, the gaps and the directions where such material shown can be used in mass production.

There some minor parts as example saying 13 MPa and 48 (page 8, line 229), its better to say 13 and 48 MPa. Please check those as there several examples in the manuscript.

Author Response

Review Report # 1

Comments and Suggestions for Authors

The manuscript "Material and environmental properties of natural polymers and their composites for packaging applications – A review" shows the use of alternative natural polymers to replace synthetic plastics. There are some good parts in the manuscript and some which needs revision. The different materials used are shown in Tables but those are not enough as the reader needs have some actual research (pictures of other work) shown how those natural plastics are processed.

Response: Thank you for your time in reviewing the manuscript and comments for improvement. The manuscript was revised based on the comments along with detailed responses. The figures (Figures 4a, 4b, and 4c) are related to the processing of polysaccharides such as chitosan, starch, and cellulose from their respective biomass resource and the commercial-scale production units are included in the revised manuscript.

  1. Please include some industrial process where natural polymers either starch, cellulose or others already applied.

Response: Thank you for your suggestion and comment. Following a brief discussion about different industrial applications of natural polymers is included on Page 18: Line number 500 of the revised manuscript.

"The protein and polysaccharides have several functional groups in their polymer structure. The hydroxyl, carboxyl, amino, and thiol functional groups of the natural polymers are modified to enhance material properties. The surface-modified natural polymers behaved like hybrid materials and showed improved adhesion, wettability, and mechanical and barrier properties and were explored to use as environment-friendly materials in many industrial applications. Because of its nontoxic, antioxidant, antibiotic, antimicrobial, and biocompatible nature, natural polymers are widely used as drug delivery, wound dressing, organ implant, and tissue engineering materials in pharmaceutical and biomedical industrial applications. The food, textiles, water treatment, and cosmetics products in the present market contain natural polymer as an emulsifier and antimicrobial and thickening agents."

  1. It would be good as the authors mention in some parts how good some materials match properties in comparison to PP and PE. Please add some sort of comparison to see where the gap that still needs to achieve replaced from synthetic to natural plastics.

Response: Thank you for your suggestion for improvement. As natural polymers have reactive functional groups, their material properties could be tailored to match the material properties of PP and PE polymers. Most of the natural polymers show comparable mechanical to consider for packaging applications. However, the water barrier properties, and water uptake. the cost-effectiveness and environmental impacts of manufacturing processes need to be improved to accelerate the commercialization of natural polymers. The current research gaps are added on page 19, line number 584 of the revised manuscript.

  1. How much of natural plastic is used and in the market? A price comparison of cost and production would be helpful.

Response: Thank you for your review comment. It is estimated that the global market for biopolymers will reach USD 10,447.2 million with a cumulative average growth rate of 17%.  The cost of biopolymers is in the range of USD 3,500 to 5,200 per ton. The market potential and cost comparison are included in the rerevised manuscript (Page 19 and 20, line number 584 onward)

  1. There are already some natural plastics used especially in India, please add some chapter to address such as it shows that progress already occur.

Response: Thank you for the suggestion to improve the manuscript. The progress in the use of biopolymers in Asian countries due to the government norms and regulations is discussed on page 20: line number 590.

  1. An outlook before the conclusion would be helpful as well to address the challenges, the gaps and the directions where such material shown can be used in mass production.

Response: Thank you for your suggestion. The challenges, research gaps, and future directions in material development with enhanced material properties, manufacturing processes with less environmental impact, and cost-effective product or material development strategies are included in the revised manuscript on pages 19 and 20 from line no 584.

  1. There are some minor parts for example saying 13 MPa and 48 (page 8, line 229), its better to say 13 and 48 MPa. Please check those as there several examples in the manuscript.

Response: Thank you for reviewing the manuscript and for your suggestion. As per the comment, the corrections are updated in the manuscript. (Page 8 - line 227 & 232)

Reviewer 2 Report

The authors did a completed review regarding natural polymers and their composite for packaging applications. The draft has a very clear and logic flow. However, here are some problems with current version:

  1. As emphasized within the title, this review paper is targeting at the discussion of packaging related applications. The authors did mention some relevant information within the draft. However, I think some further and more detailed should be included and discussed. Not only the mechanical properties but also some other relevant studies and/or applications. In other words, for the part 6, some more relevant discussion should be included.
  2. The authors did mentioned the real application relevant discussion in part 6 including the brittleness and hydrophilicity. Since it is for the real applications, stability and aging are very important factors as well. The authors should include a short discussion to cover that part.
  3. Following on the last point, the authors mentioned the good dispersion of CNC/CNF within all the natural polymers. But how about some other additives used in natural polymers? As the authors mentioned within the draft, some other additives such as cross linkers can also be used to improve the properties of natural polymers. To complete this discussion, not the compatibility of CNF/CNC but also some other additives should be discussed. 
  4. Following on last point, with that discussion, the potential phase separation/segregation and/or low compataibility caused by entropy and/or enthalpy should be discussed. There are some useful references should be included regarding the compatibility discussion: Macromolecules 2019, 52, 1526−1535; Macromolecules 2019, 52, 22, 8910–8922; Phys. Rev. Lett. 113, 225702; Macromolecules 2020, 53, 15, 6720–6730; 
  5. The table formatting within the draft is relatively low. Please try to improve it. For example, the widths of columns should be adjusted to better fit the design. For table 2, it is really difficult for authors to link the information with the corresponding references. 
  6. I noticed different vapor permeability unit was used at line 345 (g mm/ kPa/day/m2) and line 302 (g m /Pa h m2). Consistent unit should be used otherwise it is very difficult for authors to compare the results cross the studies.

Author Response

Review report # 2

Comments and Suggestions for Authors

The authors did a complete review regarding natural polymers and their composite for packaging applications. The draft has a very clear and logical flow. However, here are some problems with the current version:

Response: Thank you for your time in reviewing the manuscript and comments on improvements. The revised manuscript includes the response to each comment.

  1. As emphasized within the title, this review paper is targeting the discussion of packaging-related applications. The authors did mention some relevant information within the draft. However, I think some further and more details should be included and discussed. Not only the mechanical properties but also some other relevant studies and/or applications. In other words, for the part 6, some more relevant discussion should be included.

Response: Thank you for your comments to improve the manuscript. Natural polymers have many hydroxyl, carboxyl, amino, and thiol functional groups and are modified to enhance material properties. The surface-modified natural polymers behaved like hybrid materials and showed improved adhesion, wettability, and mechanical and barrier properties and were used as environment-friendly materials. Due to their nontoxic, antioxidant, antibiotic, antimicrobial, and biocompatible nature, natural polymers are widely used as drug delivery, wound dressing, organ implant, and tissue engineering materials in pharmaceutical and biomedical industrial applications. Part 6 of the manuscript was updated with a brief discussion on the uses of natural polymers in different industrial applications. (Page 18, Line 501 to 512)

2. The authors did mention the real application relevant discussion in part 6 including the brittleness and hydrophilicity. Since it is for real applications, stability and aging are very important factors as well. The authors should include a short discussion to cover that part.

Response: Thank you for your review and the remark on the material properties discussed in the manuscript. Natural polymers are sensitive to moisture exposure due to the presence of hydroxyl groups in their molecular structure. The studies on the surface modification for hydrophobic improvements, composite development, and crosslinking processes for the material properties improvement were discussed and section 6 was revised. (Page 19, Line 551 to 560)

3. Following the last point, the authors mentioned the good dispersion of CNC/CNF within all the natural polymers. But how about some other additives used in natural polymers? As the authors mentioned within the draft, some other additives such as crosslinkers can also be used to improve the properties of natural polymers. To complete this discussion, not the compatibility of CNF/CNC but also some other additives should be discussed. 

Response: Thank you for your review and the comment on the discussion about additives used in natural polymer studies. The natural polymers are brittle and sensitive to water contact and moisture. The plasticizer, nanofillers, and crosslinkers were used to improve the mechanical, water vapor barrier, UV barrier, and water uptake properties. A brief discussion on the additives used for the enhancement of properties is added on Page 19 line 558.

4. Following on the last point, with that discussion, the potential phase separation/segregation and/or low compatibility caused by entropy and/or enthalpy should be discussed. There are some useful references that should be included regarding the compatibility discussion: Macromolecules 2019, 52, 1526−1535; Macromolecules 2019, 52, 22, 8910–8922; Phys. Rev. Lett. 113, 225702; Macromolecules 2020, 53, 15, 6720–6730; 

Response: Thank you for your review, the comment on the discussion about phase segregation, and for sharing relevant literature. The polymer chains with low cohesive strength and more branches or rigid structures led to surface segregation during solidification and drying. A brief discussion was added, and the literature was cited in the revised manuscript. (Page 19: line 556).

5. The table formatting within the draft is relatively low. Please try to improve it. For example, the widths of columns should be adjusted to better fit the design. For table 2, it is really difficult for authors to link the information with the corresponding references. 

Response: Thank you for your review and for highlighting the table formatting issue. The width of the columns is adjusted, the justification is changed, and the grid line is introduced to improve the clarity of the presentation.

6. I noticed different vapor permeability unit was used at line 345 (g mm/ kPa/day/m2) and line 302 (g m /Pa h m2). A consistent unit should be used otherwise it is very difficult for authors to compare the results cross the studies.

Response: Thank you for your review and for highlighting inconsistency in WVP units. The values were corrected to present a consistent unit throughout the manuscript. The unit of WVP is corrected on Page 6: Table 2, Page 7: Table 3, Page 10: Line 300 and Line 303, and Page 11: line 346.

Round 2

Reviewer 2 Report

Thank you for the comments and updated draft.

It seems some pages are missing from the updated version. Please make sure the correct version is submitted for the final publication.

Except that, the authors have already addressed all my question.